# One Communication Round is All It Needs for Federated Fine-Tuning Foundation Models

## Abstract

The recent advancement of large foundation models (FMs) has increased the demand for fine-tuning these models on large-scale and cross-domain datasets. To address this, federated fine-tuning has emerged as a solution, allowing models to be fine-tuned on distributed datasets across multiple devices while ensuring data privacy. However, the substantial parameter size of FMs and the multi-round communication required by traditional federated fine-tuning algorithms result in prohibitively high communication costs, challenging the practicality of federated fine-tuning. In this paper, we are the first to reveal, both theoretically and empirically, that the traditional multi-round aggregation algorithms may not be necessary for federated fine-tuning large FMs. Our experiments reveal that a single round of communication (*i.e.,* one-shot federated fine-tuning) yields a global model performance comparable to that achieved through multiple rounds of communication. Through rigorous mathematical and empirical analyses, we demonstrate that large FMs, due to their extensive parameter sizes and pre-training on general tasks, achieve significantly lower training loss in one-shot federated fine-tuning compared to smaller models. Our extensive experiments show that one-shot federated fine-tuning not only reduces communication costs but also enables asynchronous aggregation, enhances privacy, and maintains performance consistency with multi-round federated fine-tuning for models larger than 1 billion parameters, on text generation and text-to-image generation tasks. Our findings have the potential to revolutionize federated fine-tuning in practice, enhancing efficiency, reducing costs, and expanding accessibility for large-scale models. This breakthrough paves the way for broader adoption and application of federated fine-tuning across various domains.

## 1 Introduction

Cutting-edge foundation models (FMs) demonstrate remarkable versatility across various domains. Notably, large language models (LLMs) like GPT-4 (Achiam et al., 2023), Gemma (Team et al., 2024), and Llama (Touvron et al., 2023b) excel in tasks such as translation, question answering (QA), chat assistant, and math. Similarly, stable diffusion models can generate diverse images based on textual descriptions. Achieving such versatility requires fine-tuning these FMs on cross-domain datasets. However, this process faces significant challenges in real-world scenarios due to the valuable datasets residing on devices owned by organizations or individuals, raising privacy concerns. To address these privacy issues, researchers have proposed using federated learning (FL) (Zhang et al., 2021) for distributed fine-tuning of FMs, a process known as federated fine-tuning. Federated fine-tuning allows distributed clients to collaboratively fine-tune a global FM on specific tasks without disclosing their private data.

Traditional FL requires *multiple communication rounds* between clients and the server to ensure the global model convergence (McMahan et al., 2017). However, the substantial parameter size of FMs (typically in billions) results in significant communication overhead. Many devices lack the capability to repeatedly communicate model parameters of this scale. While previous works adopt parameter-efficient fine-tuning (PEFT) methods such as low-rank adaptation (LoRA) (Hu et al., 2021) to reduce the number of trainable and communicated parameters, the high communication requirements of federated fine-tuning remain a practical limitation.

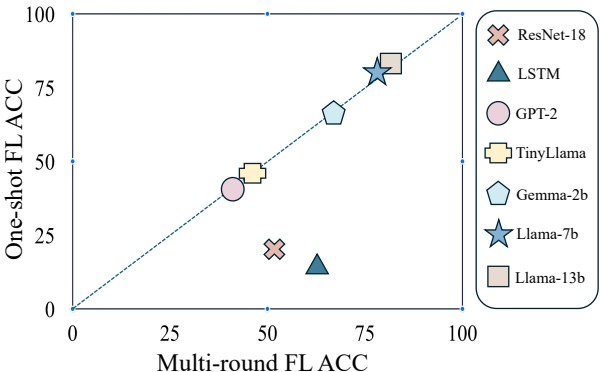

Figure 1: The distinct performances of one-shot federated learning between small models and large FMs. The horizontal axis represents multi-round FL accuracy, while the vertical axis represents one-shot FL accuracy. The ResNet-18 and LSTM are trained and tested on CIFAR-10 and Shakespeare respectively. Other models are fine-tuned on Wizard dataset and tested on ARC Easy. The closer the points are to the dashed line means the accuracy of one-shot and multi-round FL are closer in the corresponding model.

Unexpectedly, our recent experiments have discovered an emergent capability of FMs that could fundamentally shift the approach to federated fine-tuning. We find that with sufficient local fine-tuning epochs, *a single communication round is all it needs to effectively fine-tune FMs*, which is called *one-shot federated fine-tuning* (Guha et al., 2019). Figure 1 highlights the performance comparisons between one-shot FL and traditional multi-round FL, maintaining the same total number of local steps. While one-shot FL underperforms multi-round FL for smaller models (*e.g.,* ResNet-18 and LSTM), it achieves comparable performance for larger FMs (*e.g.,* GPT-2, Llama, etc). This unique discovery challenges the conventional belief that multiple communication rounds are essential for federated fine-tuning. Instead, we demonstrate that FMs can achieve convergence with just a single aggregation of well-fine-tuned local models. This paper explores this innovative finding, providing theoretical analysis and compelling empirical evidence to validate the effectiveness of one-shot FL for fine-tuning FMs.

The introduction of one-shot federated fine-tuning brings transformative benefits. First, it **dramatically reduces communication costs**. One-Shot FL slashes communication overhead by a factor of $\frac{1}{T}$, where $T$ represents the number of communication rounds in traditional federated fine-tuning. This reduction is a game-changer for devices with limited bandwidth. Second, one-shot FL enables seamless **asynchronous training**. This flexibility removes the bottleneck of server waiting times, ensuring uninterrupted training regardless of client connectivity or resource limitations. The process becomes far more robust and efficient. Third, one-shot FL offers **enhanced security** against prevalent client-side federated learning attacks. Attacks like client-side model inversion and gradient inversion, which depend on multiple global model updates, are rendered ineffective. This significantly bolsters the integrity of the training process.

Our key contributions are listed as follows:

- **Novel Discovery:** To the best of our knowledge, we are the first to discover that one communication round is sufficient for federated fine-tuning large FMs.

- **Theoretical Analysis:** We theoretically demonstrate the relationship between the global difference of one-shot federated fine-tuning and model smoothness, fine-tuning model update, number of fine-tuning rounds, and the second norm of original model parameters. Our analysis, backed by experiments, suggests that the lower difference between one-shot and multi-round federated fine-tuning of larger FMs is related to the inherent smoothness of FMs and the smaller parameter updates during fine-tuning.

- **Experimental Validation:** We conduct extensive experiments on six FMs and three tasks, demonstrating that one-shot federated fine-tuning achieves performance comparable to multi-round federated fine-tuning, particularly for models with over 1 billion parameters. Experimental results also surprisingly show that LoRA outperforms full fine-tuning in the context of one-shot federated fine-tuning.

## 2 PRELIMINARY

**Federated Learning Paradigm of Small Models.** In FL, the primary objective is to optimize a global objective function $F(\boldsymbol{w})$, which is weighted average of the local objective functions from $m$ clients (Wang et al., 2020b):

$$F(\boldsymbol{w}) = \sum_{i=1}^{m} p_i F_i(\boldsymbol{w}) \tag{1}$$

where $\boldsymbol{w}$ represents the model parameters and $p_i$ is the scaling factor for averaging. To protect the data privacy of each client, the server cannot access the local dataset. Thus, the local objective function $F_i(\boldsymbol{w})$ remains unknown to the server. FedAvg (McMahan et al., 2017) algorithm provides a distributed training algorithm to facilitate privacy-conscious training. It allows multiple clients to train the model on their local datasets and aggregates locally trained models on the server at the end of each communication round. In $t$-th communication round, the global model update rule of FedAvg is:

$$\boldsymbol{w}^{(t+1,0)} - \boldsymbol{w}^{(t,0)} = \alpha^{(t)} \sum_{i=1}^{m} p_i \Delta_i^{(t)}, t \in [0, T-1] \tag{2}$$

where $\boldsymbol{w}^{(t,0)}$ is the model weights in $t$-th communication round and 0-th local step, which represents the global model in $t$-th round. $T$ is the total number of communication rounds, $\alpha^{(t)}$ is the global learning rate, and $\Delta_i^{(t)}$ is the local model update in $t$-th round. $\Delta_i^{(t)}$ is the accumulative model update of $k$ local stochastic gradient descent (SGD) steps:

$$\Delta_i^{(t)} = \sum_{j=1}^{k} \beta_i^{(t,j)} g_i(\boldsymbol{w}_i^{(t,j)}) \tag{3}$$

where $g_i(\boldsymbol{w}_i^{(t,j)})$ is the stochastic gradient over a local mini-batch and $\beta_i^{(t,j)}$ is the local learning rate. Note that $j$ here represents a mini-batch, and $k$ is the total number of mini-batches per client.

Local datasets in FL are typically heterogeneous, leading to differences in local objectives. Therefore, FL usually converges more slowly than centralized machine learning. This slow convergence necessitates a large number of global communication rounds and local steps to achieve satisfactory performance. For example, experiment results in (Reddi et al., 2020) show that the ResNet-18 model requires more than 2000 and 4000 communication rounds to converge on CIFAR-10 (Krizhevsky et al., 2009) and CIFAR-100 respectively. Even for simple natural language processing tasks such as Shakespeare, an RNN model needs more than 50 rounds to converge. The requirement for multi-round communication rounds introduces several significant drawbacks. First, clients must frequently exchange model parameters with the server, which can be prohibitively expensive in certain constrained scenarios or on devices with limited resources. Second, repeated invocation of computational resources for training increases the overall computational overhead. Additionally, the multi-round communication approach leads to excessive energy consumption, synchronization difficulties, and challenges in maintaining privacy protection. Thus, optimizing FL algorithms to minimize the number of communication rounds is an essential research direction in FL.

**Federated Fine-Tuning Foundation Models.** Foundation models (FMs) (Zhou et al., 2023) refer to pre-trained deep learning models with a vast number of parameters, typically in the order of billions. These FMs are trained on broad data at scale and are adaptable to a wide range of downstream tasks when fine-tuned on domain-specific datasets (Bommasani et al., 2021). Since domain-specific datasets are often distributed across multiple devices, FL offers an important paradigm for fine-tuning FMs while preserving data privacy.

Federated fine-tuning also adopts the same FedAvg algorithm in Eq. 1 and Eq. 2 to aggregate the local model updates. The key difference lies in the *model parameter size*. The parameter size of large FMs is usually hundreds of times greater than that of small models, resulting in a significant increase in the computation resources and communication overhead required for federated fine-tuning. Given the network communication capabilities of commonly used devices, performing multi-round synchronized communication of large model parameters between servers and clients is virtually impossible. Although parameter-efficient fine-tuning algorithms like LoRA (Hu et al., 2021) have been adopted, the communication overhead remains excessively high, hindering practical application.

**One-Shot Federated Learning.** To reduce communication overhead in FL, recent works have focused on one-shot FL (Jhunjhunwala et al., 2024; Guha et al., 2019; Gong et al., 2021; Li et al., 2020; Zhou et al., 2020; Yang et al., 2024), which uses a single communication round to obtain the global model. These algorithms often employ knowledge distillation or neuron-matching methods to optimize the global model. However, these approaches require additional data or computation. Knowledge distillation often necessitates auxiliary public datasets or external generative models, and neuron matching requires additional computation on both clients and the server. Despite these additional resource requirements, the performance of one-shot FL has historically been inferior to standard multi-round FL. For instance, experiments in (Jhunjhunwala et al., 2024) show that one-shot FL achieves only 50% accuracy on the CIFAR-10 dataset, which is 20% lower than the accuracy achieved with 5-round FL.

However, our recent experiments have uncovered greater potential for one-shot FL in federated fine-tuning large FMs. As shown in Figure 1, one-shot FL for large models does not show a significant performance gap compared to multi-round FL, which is commonly observed with smaller models. In fact, when the total number of local steps is the same, the performance of large models fine-tuned by one-shot FL is comparable to that of multi-round FL. Additionally, in fine-tuning larger models such as Llama-13b, one-shot FL even performed slightly better than multi-round FL. These results, along with the experiment results in Section 4, suggest that traditional multi-round FL algorithms like FedAvg may no longer be necessary for federated fine-tuning large FMs. Large FMs can effectively learn downstream tasks from distributed clients with just a single communication round, opening up new possibilities for federated fine-tuning applications.

Although we have observed consistently good performance with one-shot federated fine-tuning, the reasons for the divergent results between one-shot FL in fine-tuning large models and training small models remain unexplored. In the next section, we will delve into this phenomenon through theoretical analysis.

## 3 THEORETICAL ANALYSIS OF ONE-SHOT FEDERATED FINE-TUNING

For a multi-round FL algorithm, if the total number of communication rounds is $T$ and the number of local SGD steps for each round is $k$, according to Eq. 2 the global model parameters after FL satisfy:

$$\boldsymbol{w}^{(T,0)} - \boldsymbol{w}^{(0,0)} = \sum_{t=0}^{T-1} \alpha^{(t)} \sum_{i=1}^{m} p_i \Delta_i^{(t)}, \tag{4}$$

where $\Delta_i^{(t)}$ is defined by Eq. 3. For a specific client $i$, the *accumulated* local model update $\Delta_i$ is:

$$\Delta_i = \sum_{t=0}^{T-1} \Delta_i^{(t)} = \sum_{t=0}^{T-1} \sum_{j=1}^{k} \beta_i^{(t,j)} g_i(\boldsymbol{w}_i^{(t,j)}), \tag{5}$$

In contrast, for one-shot FL with $T = 1$, the accumulated local model update is:

$$\Delta_i = \sum_{j=1}^{Tk} \beta_i^{(0,j)} g_i(\boldsymbol{w}_i^{(0,j)}), \tag{6}$$

Note that one-shot FL doesn't mean that we only train the model for one epoch. We set the number of steps per client to $Tk$ since we are trying to match the total number of epochs with the multi-round FL. The reason why the one-shot FL performs differently from the multi-round FL lies in the difference between the local model updates in Eq. 5 and Eq. 6. In Eq. 5, after the $t$-th communication round, the local training starts from the *updated* global model parameter $\boldsymbol{w}^{(t,0)}$ sent by the server, which is aggregated from all the local model updates in $t$-th round and contains richer local knowledge. On the contrary, in one-shot FL (Eq. 6), even with the same total local steps, clients can only continuously train the local models *without* global information. The poor performance of one-shot FL is due to the gradients calculated on the local models being less accurate than those calculated on the aggregated global model. This can be expressed in mathematical form:

$$\varepsilon_i = \sum_{j=k+1}^{Tk} \beta_i^{(0,j)} [(g_i(\boldsymbol{w}_i^{(0,j)}) - g_i(\boldsymbol{w}_i^{(t,j-kt)}))], \quad \text{where } t = \lceil \frac{j}{k} \rceil \tag{7}$$

where $\lceil \cdot \rceil$ means ceiling. $\varepsilon_i$ is the accumulated local update difference between one-shot FL and multi-round FL. Since the global model is aggregated by local models, the global model update difference $\varepsilon_i$ is the aggregation of local differences. Its L2 norm can then be bounded by the sum of local differences with triangle inequality, which is:

$$\|\varepsilon\| \leq \sum_{i=1}^{m} \|\varepsilon_i\|, \tag{8}$$

The global difference can be further simplified by the following assumptions.

**Assumption 1 (Model Smoothness).** The objective function of the pre-trained large FM is Lipschitz smooth with an $L$ value, that is $\|\nabla F_i(\boldsymbol{w}_x) - \nabla F_i(\boldsymbol{w}_y)\| \leq L\|\boldsymbol{w}_x - \boldsymbol{w}_y\|$, $L > 0$, where $\nabla F_i(\cdot)$ is the model gradient.

**Assumption 2 (Bounded Model Updates).** The model updates during FL are much smaller than the initial model parameters, that is, $\|\boldsymbol{w}^{(t,j)} - \boldsymbol{w}^{(0,0)}\| \leq \tau\|\boldsymbol{w}^{(0,0)}\|$, $0 < \tau < 1$.

**Theorem 1 (The one-shot global difference is related to $L$, $\tau$, epoch numbers $Tk$, and number of clients $m$).** Under Assumptions 1 and 2, ignoring the learning rates, the difference between one-shot FL and multi-round FL can be bounded as follows:

$$\|\varepsilon\| \leq \Gamma\|\boldsymbol{w}^{(0,0)}\|, \text{ where } \Gamma = L\tau Tkm \tag{9}$$

This equation indicates that with lower values of $L$, $\tau$, $T$, $k$, and $m$, the model update of one-shot FL will be closer to that of multi-round FL. Since our experiments have shown that LLMs exhibit significant advantages over small models in one-shot learning, we conduct experiments on the factors in Equation 9 to provide a detailed explanation of this phenomenon.

**Foundation Models are Extremely Smooth ($L_{FM} \ll 1$).** In Equation 9, the factor $L$ represents the smoothness of the model, with smaller $L$ implying a smoother model. We argue that pre-trained large FMs are much smoother than small models and thus have much smaller $L$ values. Large FMs are pre-trained on large-scale datasets to obtain general capabilities. During this pre-training process, the parameters of FMs are optimized from the ridges to the basins in the loss landscape. Additionally, as observed in a previous work (Ainsworth et al., 2022), wider models have more flattened basins in the loss landscapes. With these pieces of prior knowledge, we hold the contention that the loss landscape in large FM fine-tuning is much **flatter** and **smoother** than that in training small models from scratch, resulting in much smaller $L$ values. To verify this argument, we estimate $L$ by $L = \frac{\|\nabla F_i(\boldsymbol{w}_x) - \nabla F_i(\boldsymbol{w}_y)\|}{\|\boldsymbol{w}_x - \boldsymbol{w}_y\|}$. We randomly sample a mini-batch of data in the training datasets and compute the gradient on $\boldsymbol{w}^{(0,0)}$ and $\boldsymbol{w}^{(T,k)}$ to get $\nabla F_i(\boldsymbol{w}^{(0,0)})$ and $\nabla F_i(\boldsymbol{w}^{(T,k)})$. Then we visualize the value of $\frac{\|\nabla F_i(\boldsymbol{w}^{(0,0)}) - \nabla F_i(\boldsymbol{w}^{(T,k)})\|}{\|\boldsymbol{w}^{(0,0)} - \boldsymbol{w}^{(T,k)}\|}$ in Figure 2(a). According to Figure 2(a), FMs (*i.e.,* models to the right of the red dash line) have much smaller $L$ values than small models, which is consistent with our conjecture.

**Foundation Models Have Much Smaller Model Updates in Fine-Tuning ($\tau_{FM} \ll 1$).** Another crucial distinction in our analysis lies in the different tasks in FL: **fine-tuning** and **training from scratch**. Since the fine-tuning task updates the model parameters to adapt to downstream tasks without compromising its performance on the general task, it will only slightly update the model parameters. Therefore, we argue that the model parameter updates in the fine-tuning process are much smaller than the pre-trained model parameters, *i.e.,* $\|\boldsymbol{w}^{(t,j)} - \boldsymbol{w}^{(0,0)}\| \ll \|\boldsymbol{w}^{(0,0)}\|$. In this case, the federated fine-tuning task would have a very small $\tau$ in Equation 9. To verify this, we conduct experiments to estimate the $\tau$ values by $\frac{\|\boldsymbol{w}^{(T,k)} - \boldsymbol{w}^{(0,0)}\|}{\|\boldsymbol{w}^{(0,0)}\|}$, where $\boldsymbol{w}^{(T,k)}$ represents the model update after the entire fine-tuning process on the training datasets. We visualize the estimated $\tau$ values of different models in Figure 2(b), which illustrates that the $\tau$ values in FMs are much smaller than those in small models.

**Large Foundation Models Require Less Fine-Tuning Steps ($Tk_{FM} \ll Tk_{small}$).** Different from training a small model from scratch, fine-tuning a large model typically doesn't require a large number of total training steps to ensure convergence. This is mainly because the pre-trained models will be overfitting on the fine-tuning data with too many epochs, which will destroy the model's ability on the general tasks. As a result, the $Tk$ values of large FMs are also smaller than those in small models. Table. 1 displays the $T$ and $k$ numbers adopted by our experiments.

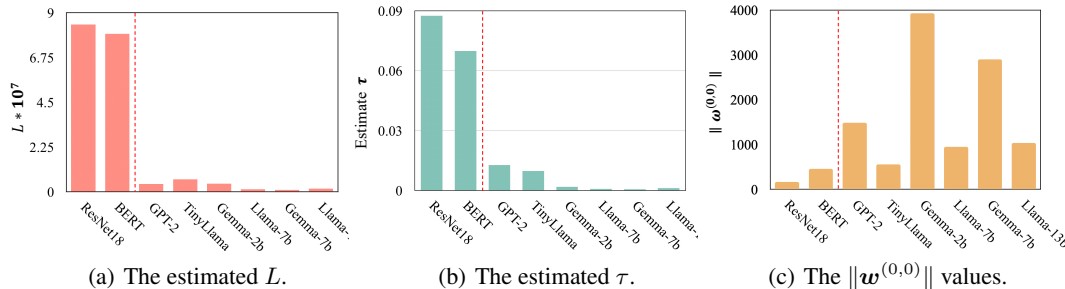

(a) The estimated $L$.      (b) The estimated $\tau$.      (c) The $\|\boldsymbol{w}^{(0,0)}\|$ values.

Figure 2: Experiment results on $L$, $\tau$, and $\|\boldsymbol{w}^{(0,0)}\|$ in different models. We use the CIFAR-10 dataset to compute the gradient on ResNet18 (He et al., 2016). We use the WizardLM dataset (Xu et al., 2023) to compute the gradient on the language models. Models to the left of the red dashed line are small models, while those to the right are foundation models (FMs). The figures indicate that large FMs have significantly smaller $L$ and $\tau$ values compared to small models. Additionally, $\|\boldsymbol{w}^{(0,0)}\|$ does not increase proportionally with the model size. In conclusion, without considering other unrelated influencing factors, the value of $\Gamma\|\boldsymbol{w}^{(0,0)}\|$ decreases as the model size increases.

Table 1: $Tk$ settings in experiments. $T$ is the number of global communication rounds. $k$ is the total number of local SGD steps, which is computed by (dataset length $\times$ epoch number / batch size).

|  | ResNet-18 | BERT | GPT-2 | TinyLlama | Gemma-2b | Llama-7b | Gemma-7b | Llama-13b |
|---|---|---|---|---|---|---|---|---|
| **T** | 50 | 50 | 5 | 3 | 3 | 3 | 3 | 3 |
| **k** | 7812 | 3906 | 5625 | 3750 | 1875 | 1875 | 1875 | 1875 |
| **Tk** | 390600 | 195300 | 28125 | 11250 | 5625 | 5625 | 5625 | 5625 |

We also visualize $\|\boldsymbol{w}^{(0,0)}\|$ in Figure 2(c). Although the $\|\boldsymbol{w}^{(0,0)}\|$ value of the small model is relatively small, it does not exhibit a clear trend positively correlated with model size (*e.g.,* TinyLlama has a similar $\|\boldsymbol{w}^{(0,0)}\|$ value with BERT, but has 10 times more parameters than BERT, Gemma-2b has much larger $\|\boldsymbol{w}^{(0,0)}\|$ value than Llama-13b).

**Conclusion: Large Foundation Models Have Smaller Global Model Update Difference $\varepsilon$.** Based on the discussion before regarding the $L$, $\tau$, $Tk$, and $\|\boldsymbol{w}^{(0,0)}\|$ values of the model with various sizes, we conclude that large FMs have smaller $L$, $\tau$, and $Tk$ values, while $\|\boldsymbol{w}^{(0,0)}\|$ is not strongly related to the model size. We ignore the client number $m$ and visualize the $\|\varepsilon\| = \Gamma\|\boldsymbol{w}^{(0,0)}\|$ values of different models in Figure 3. The results in Figure 3 clearly demonstrate that large FMs (GPT-2 and all models to its right) have significantly lower $\|\varepsilon\|$ values than the small models, with larger FMs having lower values. According to Eq. 9, smaller $\|\varepsilon\|$ means a smaller difference between one-shot and multi-round FL. Consequently, FMs have much better one-shot FL performance than small models.

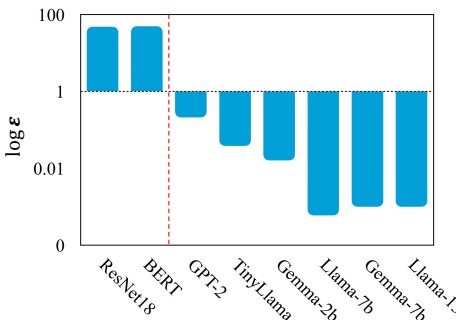

Figure 3: The estimated $\log\|\varepsilon\|$ in different models calculated by $\log\|\varepsilon\| = \log(L\tau Tk\|\boldsymbol{w}^{(0,0)}\|)$.

In summary, the reasons why large FMs have smaller differences in one-shot federated fine-tuning are due to three main factors. ***Fisrt, the pre-trained FMs have extremely smooth loss landscapes in fine-tuning, i.e., $L_{FM} \ll 1$. Second, the fine-tuning model updates are particularly small compared to the pre-trained parameters, i.e., $\tau_{FM} \ll 1$. Third, FM fine-tuning requires far fewer epochs than training small models from scratch, i.e., $Tk_{FM} \ll Tk_{small}$. These three factors lead to much smaller error $\varepsilon$ in the one-shot federated fine-tuning of FMs.***

Table 2: Performance of one-shot federated fine-tuning in Q&A tasks. The rows with star (*) are the results of one-shot federated fine-tuning.

| Tasks | Methods | TinyLlama | | | Gemma-2b | | | Llama-7b | | | Llama-13b | | |
|---|---|---|---|---|---|---|---|---|---|---|---|---|---|
| | | MMLU | Wizard | M-W | MMLU | Wizard | M-W | MMLU | Wizard | M-W | MMLU | Wizard | M-W |
| MMLU | LoRA | **25.08** | **25.07** | 24.98 | **38.43** | **37.75** | **37.69** | **36.16** | 35.07 | **35.37** | 47.22 | 46.83 | 46.82 |
| | **LoRA*** | 25.01 | 25.04 | **25.03** | 38.24 | 36.55 | 35.14 | 35.86 | **35.91** | 34.84 | **48.40** | **47.93** | **47.43** |
| | Full FT | **27.30** | 24.84 | **25.46** | **42.02** | **34.60** | 28.36 | **45.61** | 30.52 | 28.81 | **50.24** | **42.12** | **32.91** |
| | **Full FT*** | 26.39 | **24.87** | 24.99 | 40.93 | 33.86 | **28.71** | 44.20 | **33.97** | **29.05** | 48.30 | 39.62 | 29.76 |
| ARC | LoRA | 35.49 | **37.28** | **36.69** | **43.09** | 43.26 | 42.06 | 50.43 | 50.94 | 51.19 | 55.72 | 55.72 | 55.63 |
| | **LoRA*** | **36.86** | 36.77 | 36.26 | 40.61 | 42.49 | **42.15** | **50.85** | **51.88** | **52.13** | **56.40** | **58.11** | **56.74** |
| | Full FT | 32.76 | **37.03** | 33.02 | **41.04** | 45.48 | **37.46** | **43.26** | 40.24 | **37.15** | 42.41 | **47.57** | **42.75** |
| | **Full FT*** | **33.19** | 36.26 | **33.87** | 39.85 | **45.92** | 34.47 | 41.72 | **43.52** | 37.03 | **44.62** | 45.05 | 40.21 |

# 4 EXPERIMENT

## 4.1 EXPERIMENTAL SETUPS

**Models and Datasets.** To demonstrate the performance of FMs of different sizes on one-shot federated fine-tuning, we selected multiple models ranging in parameter size from 1b to 13b for experiments. The language models we experimented with range in parameter size from smallest to largest as follows: TinyLlama (1.1b) (Zhang et al., 2024b), Gemma-2b (Team et al., 2024), Llama-7b, and Llama-13b (Touvron et al., 2023a). We use the MMLU (Hendrycks et al., 2020) training dataset and Wizard (Luo et al., 2023) dataset to federated fine-tune these models. For evaluation, we leverage MMLU and ARC Challenge (Clark et al., 2018) in Eval-Harness (Gao et al., 2023) to evaluate the model ability of QA tasks, and the GPT-4 evaluation in MT-bench (Zheng et al., 2023) for the chat assistant task.

**Federated Fine-Tuning Settings.** For federated fine-tuning on a single MMLU or Wizard dataset, we randomly split the dataset into 10 clients. We also have a strongly non-iid setting, which assigns the MMLU dataset to 10 clients and the Wizard dataset to another 10 clients, and lets the 20 clients fine-tune the FM. For the baseline, we use a multi-round FedAvg algorithm on both LoRA and full fine-tuning. For our one-shot federated fine-tuning, we only perform a single communication round. To ensure fairness, we keep the total number of local steps the same between multi-round and one-shot federated fine-tuning. *e.g.,* , if the setting in multi-round federated fine-tuning is 3 communication rounds, 1 local epoch in each round, the setting in one-shot should be 1 communication round, 3 local epochs in that round.

## 4.2 MAIN RESULTS

**One-Shot Federated Fine-Tuning in QA Tasks.** We first evaluate the performance of one-shot federated fine-tuning in QA tasks and display the results in Table 2. The columns with titles MMLU, Wizard, and M-W represent the model fine-tuned by MMLU, Wizard, and the mixture of MMLU and Wizard datasets respectively. The rows with the title MMLU and ARC represent the model accuracy evaluated by the MMLU test set and ARC Challenge. The Methods columns mean the fine-tuning is performed by LoRA or full fine-tuning, while the rows with a star (*) represent one-shot federated fine-tuning. According to Table 2, the performance of one-shot federated fine-tuning is generally comparable to that of multi-round federated fine-tuning. In some settings, one-shot fine-tuning even achieves higher accuracy. For example, the Llama-13b model one-shot fine-tuned by LoRA on the Wizard dataset achieves 47.93% accuracy on MMLU and 58.11% on ARC Challenge, which is higher than the 46.83% and 55.72% accuracy of multi-round fine-tuning. In full fine-tuning, multi-round fine-tuning performs better in some settings. For instance, the Llama-13b model multi-round full fine-tuned on the Wizard dataset outperforms one-shot fine-tuning on both MMLU and ARC Challenge. These observations align with our previous theoretical analysis. Full fine-tuning involves greater parameter updates compared to LoRA, resulting in a larger $\tau$ value, and thus a larger $\varepsilon$ value. Consequently, the performance of one-shot full fine-tuning may sometimes be inferior to LoRA fine-tuning. However, this does not affect our overall conclusion: **for large FMs, one-shot federated fine-tuning can effectively replace multi-round federated fine-tuning.** One-shot fine-tuning provides comparable performance to multi-round fine-tuning while significantly reducing communication costs.

| Models | Methods | MMLU | Wizard | M-W | AVG. | Base |
|---|---|---|---|---|---|---|
| **TinyLlama** | LoRA | 3.59 | 3.44 | 3.65 | **3.56** | |
| | **LoRA\*** | 3.33 | 3.45 | 3.74 | 3.51 | 3.47 |
| | Full FT | 2.02 | 3.76 | 2.97 | **2.92** | |
| | **Full FT\*** | 1.91 | 4.21 | 2.38 | 2.83 | |
| **Gemma-2b** | LoRA | 3.36 | 3.48 | 3.46 | 3.43 | |
| | **LoRA\*** | 3.23 | 3.77 | 3.66 | **3.55** | 3.60 |
| | Full FT | 2.16 | 4.36 | 2.75 | **3.09** | |
| | **Full FT\*** | 1.92 | 4.27 | 2.50 | 2.90 | |
| **Llama-7b** | LoRA | 3.01 | 3.27 | 2.99 | 3.09 | |
| | **LoRA\*** | 2.69 | 3.90 | 3.54 | **3.38** | 2.86 |
| | Full FT | 1.85 | 4.18 | 2.31 | 2.78 | |
| | **Full FT\*** | 1.56 | 4.79 | 2.21 | **2.85** | |
| **Llama-13b** | LoRA | 2.58 | 2.68 | 2.86 | 2.71 | |
| | **LoRA\*** | 3.02 | 4.27 | 3.26 | **3.52** | 2.69 |
| | Full FT | 2.43 | 4.63 | 3.05 | **3.37** | |
| | **Full FT\*** | 1.81 | 4.74 | 2.62 | 3.06 | |

Table 3: Performance of one-shot federated fine-tuning on chat assistant tasks. Wizard has better performance than MMLU on MT-bench. We use AVG. column to show the averaging performance of specific methods.

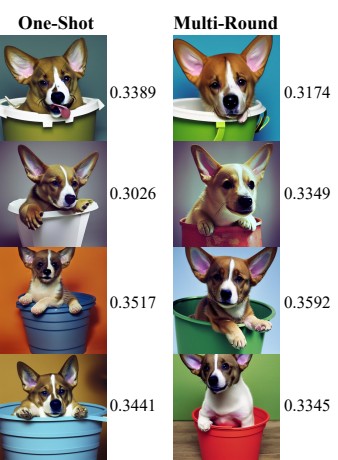

Figure 4: "A photo of a dog in a bucket" generated by LoRA fine-tuned stable diffusion models.

**One-Shot Federated Fine-Tuning in Chat Assistant Tasks.** We evaluate the performance of FMs in chat assistant tasks, where models generate answers to several questions and are scored by GPT-4. The score from MT-bench is the average score across all questions. Table 3 shows the scores of multi-round and one-shot federated fine-tuned models. The averaging scores of three fine-tuning datasets indicate that larger FMs perform better in one-shot federated fine-tuning. Specifically, multi-round fine-tuning outperforms one-shot fine-tuning in both LoRA and full fine-tuning on the Tinyllama model, which is the smallest model in our experiments. On the contrary, for larger models, such as Gemma-7b and Llama-13b, one-shot fine-tuning performs better than multi-round fine-tuning. This observation aligns with our previous theoretical analysis that larger models have smaller differences between one-shot and multi-round FL. The superior performance of one-shot fine-tuning in larger models might be attributed to the less number of local epochs per round, which leads to a slower local learning rate decay. The chat assistant's capabilities may benefit from this smoother learning rate decay process.

**One-Shot Federated Fine-Tuning in Text-To-Image Generation Tasks.** In addition to testing LLMs, we also evaluated the effectiveness of one-shot federated fine-tuning in the text-to-image generation tasks. We use LoRA to fine-tune a stable-diffusion-v1-5 (Rombach et al., 2022) model on the Dreambooth (Ruiz et al., 2023) dataset with 5 distributed clients. In the multi-round setting, we have 5 global rounds, with 5 local epochs in each round. In the one-shot setting, we have 1 global round and 25 local epochs in that round. After fine-tuning, we evaluated the models using the CLIP (Hessel et al., 2021) score with ViT-B-32 (Dosovitskiy et al., 2020) to assess the quality of generated images based on specific prompts. Figure 4 shows the images generated with the prompt "A photo of a dog in a bucket" The right column displays the result of multi-round federated fine-tuning, while the left column shows the result from the one-shot setting. The numbers to the right of the images represent the CLIP scores. The qualities of the images generated by both methods are essentially the same. The average CLIP score in the one-shot setting is 0.3343, while the score in the multi-round setting is 0.3345. These results indicate that the effectiveness of one-shot federated fine-tuning extends to fine-tuning stable diffusion models.

## 5 DISCUSSION

**One-Shot Federated Fine-Tuning Saves Communication Cost.** In FL, the server needs to send the model parameters to all the selected clients and receive the model updates from the clients in each communication round. Thus, the total number of communicated parameters in multi-round should be $2mTS$, where $S$ is the model size. In one-shot federated fine-tuning, the server and the clients only perform one-round communication, so the number of communicated parameters is only $2mS$. This reduction in communication overhead is significant, especially in the federated fine-tuning of large

FMs. For instance, the Llama-13b model has approximately 50GB parameters, *i.e.,* $S = 50GB$. In our experiments, the 3-round federated fine-tuning on Llama-13b needs to communicate 3000GB data between the server and the clients, which may be unaffordable in scenarios with tight communication budgets. However, one-shot federated fine-tuning reduces this amount to 1000GB. This substantial reduction in communication makes federated fine-tuning of large FMs more practical and affordable in real-life scenarios.

**One-Shot Federated Fine-Tuning Supports Asynchronous Global Aggregation.** In traditional multi-round FL, clients need to train local models synchronously. The server can only perform the aggregation and send the new global model to clients after receiving all local model updates. This requirement poses challenges for federated learning applications. For example, if local computation resources are occupied by other tasks or if the connection between the server and clients is unstable, the training process will be halted. One-shot federated fine-tuning effectively addresses this problem. The server can update the global model with local updates as soon as they are received, allowing for real-time model updates. Therefore, even if some clients fail to send model updates promptly due to various reasons, the global model on the

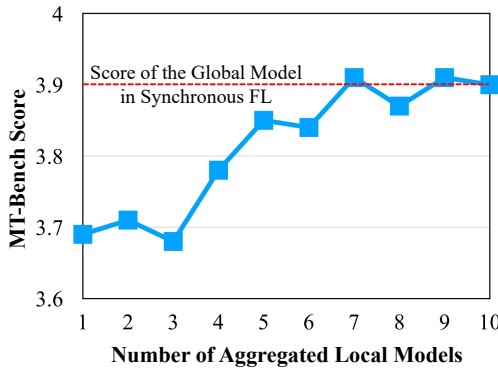

Figure 5: The MT-bench score of the global model merged by a varied number of clients.

server can still be updated by most clients, resulting in a usable global model. To further illustrate this point, we sequentially aggregated local model updates from client 1 to client 10 in one-shot federated fine-tuning of Llama-7b on the Wizard dataset. We tested the global model's performance on the MT-bench as we aggregated updates from 1, 2, 3, ..., and up to 10 clients. The results are displayed in Figure 5. The model score increases as more clients contribute their local updates to the global model, indicating that each individual local model update provides an immediate improvement in global model performance. The red dash line represents the model score in the synchronous FL setting, which is equal to the score of aggregating ten clients in asynchronous FL.

**One-Shot Federated Fine-Tuning Naturally Mitigates Client-Side Privacy Threatens.** In a traditional FL algorithm, clients repeatedly receive new global model parameters each round, which could lead to client-side privacy issues. Malicious clients can exploit model inversion (Fredrikson et al., 2015; Zhang et al., 2020) and gradient inversion attacks (Huang et al., 2021) to recover private training samples or user inputs from other clients (Wei et al., 2023). These attacks heavily rely on access to the global model parameters and certain data distribution information. However, in one-shot FL, the server can choose not to send back global parameters to the clients and only provide an API of the fine-tuned model. By doing this, it can eliminate the possibility of client-side privacy attacks.

## 6 CONCLUSION

In this paper, we tackle the critical issue of high communication costs that limit the practical application of federated fine-tuning. Through a series of experiments, we demonstrate that multi-round communication is not necessary for fine-tuning FMs, as one-shot federated fine-tuning achieves comparable performance. We then provide a theoretical analysis to explain why one-shot federated fine-tuning is effective for large FMs and validate our findings with empirical evidence. Our extensive experiments show that one-shot federated fine-tuning performs on par with multi-round federated fine-tuning across 5 different models and 3 diverse tasks. This method significantly reduces communication overhead, making federated fine-tuning more feasible and efficient, especially for large-scale models. Moreover, one-shot federated fine-tuning supports asynchronous local updates and enhances security by minimizing data exposure during the training process. These findings make it possible to harness the power of large FMs in environments with limited communication resources, thereby broadening the accessibility and utility of advanced AI technologies.

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

## A  RELATED WORK

**One-Shot Federated Learning.**  One-shot federated learning refers to learning the parameters of the global model in a single round of communication between clients and the server (Guha et al., 2019). There are two main strategies for optimizing one-shot FL, neuron matching and knowledge distillation. Neuron matching is based on the permutation symmetry of neural networks (Ainsworth et al., 2022), which means that client model parameters can be aligned according to a common ordering and then be averaged. Previous works use algorithms such as the Fisher information matrix (Jhunjhunwala et al., 2024) and permutation matrix (Wang et al., 2020a) to match the local model parameters. The knowledge distillation methods aim at distilling knowledge from well-trained local models through public data (Gong et al., 2021; Li et al., 2020; Heinbaugh et al., 2022). Some works also use distilled data to transfer knowledge between clients and the server (Zhou et al., 2020). Recent works adopt generative models to help generate substitute data for the local dataset on the server (Yang et al., 2024; Zhang et al., 2022).

**Federated Fine-Tuning.**  Federated fine-tuning (Orescanin et al., 2021; Cheng et al., 2021) aims to fine-tune FMs by cross-domain on-device datasets while preserving data privacy. Recent works use PEFT methods such as LoRA (Hu et al., 2021) in federated fine-tuning (Zhang et al., 2024a) to save communication and computation costs. Federated fine-tuning also faces similar research problems as FL. Current works have discussed the non-IID problem (Cho et al., 2024) and personalized federated fine-tuning (Wagner et al., 2024).

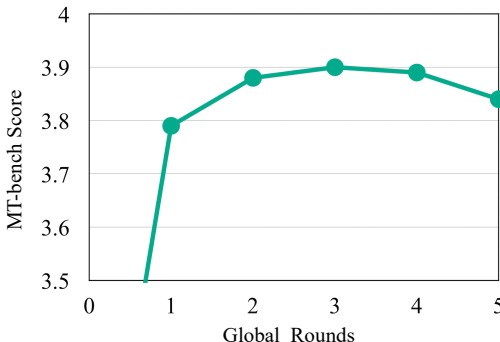

Figure 6: The MT-bench score of global model in 1-5 global rounds.

## B    ADDITIONAL EXPERIMENTAL SETUPS

**Computer Resources.**    We used a 256GB AMD EPYC 7763 64-Core Processor on Linux v4.18.0 to run the experiments. For LoRA fine-tuning on all the models and full fine-tuning on all the models except Llama-13b, we used 4 NVIDIA RTX A6000 GPUs. For Llama-13b full fine-tuning, we use 8 NVIDIA A100 GPUs.

**Hyperparameter Settings.**    For LoRA fine-tuning across all the models and datasets, we set the local LoRA rank to 16, the local learning rate to 3e-4, and the batch size to 64. For full fine-tuning, we reduced the learning rate to 3e-5 and set the learning rate to 8. For multi-round settings, the numbers of global communication rounds and local epochs in each round in different models and datasets are listed in Table 4. The one-shot setting satisfies $T = 1$ and $k$ equals $Tk$ in the multi-round setting. The number of rounds and epochs we selected can ensure convergence and avoid overfitting. We show a simple example in Appendix C to demonstrate this point.

Table 4: Global rounds and local epochs settings in multi-round experiments.

| Models | TinyLlama | | | Gemma-2b | | | Llama-7b | | | Llama-13b | | |
| | MMLU | Wizard | M-W | MMLU | Wizard | M-W | MMLU | Wizard | M-W | MMLU | Wizard | M-W |
| --- | --- | --- | --- | --- | --- | --- | --- | --- | --- | --- | --- | --- |
| T | 3 | 3 | 3 | 3 | 3 | 3 | 3 | 3 | 3 | 3 | 3 | 3 |
| k | 1 | 2 | 1 | 1 | 2 | 1 | 2 | 1 | 1 | 1 | 1 | 1 |

## C    ADDITIONAL EXPERIMENTAL RESULTS

**Zero-Shot Results.**    We test the zero-shot performance of models used in Table 2 for reference. The results are displayed in Table 5

**Standalone Results of Local Models.**    To further demonstrate the effectiveness of federated fine-tuning, we performed the standalone experiment to compare the performance of the global model and the local model only trained on local datasets. We did the experiments on the llama-7b model and Wizard dataset and displayed the results in Table 6. The results show that the accuracy of most local models is slightly lower than that of the global model, with some local models outperforming the global model. This is reasonable in the context of the federated fine-tuning task because the models have already been pre-trained. Therefore, even though clients have less training data, the performance of local models does not differ significantly from the global model.

**More Global Round Settings.**    We also tested the model performance when we had more and fewer global rounds in a multi-round setting. We evaluated the global model in 1, 2, 3, 4, and 5 global rounds when fine-tuning the Llama-7b model on Wizard dataset. The results are shown in Figure 6. In the first round, the MT-bench score increases from the 2.86 in base model to around 3.80. Then, it slightly increases towards 3.90 in the 3rd round and begins to decrease afterward. A similar phenomenon can be seen in other datasets and models that the model performance will increase in

Table 5: Zero-Shot results of models on MMLU and ARC Challenge.

| Tasks | TinyLlama | Gemma-2b | Llama-7b | Llama-13b |
|-------|-----------|----------|----------|-----------|
| MMLU  | 24.90     | 34.63    | 34.44    | 46.23     |
| ARC   | 35.41     | 40.25    | 45.65    | 51.79     |

Table 6: Standalone results of 3-epochs federated fine-tuning on Llama-7b with Wizard dataset. The numeric header columns indicate the ARC Challenge accuracy of the local models only fine-tuned on their local dataset for 3 epochs.

| One-Shot | 0 | 1 | 2 | 3 | 4 | 5 | 6 | 7 | 8 | 9 |
|----------|-------|-------|-------|-------|-------|-------|-------|-------|-------|-------|
| 51.88 | 50.79 | 51.02 | 50.05 | 50.43 | 52.33 | 51.22 | 51.28 | 52.30 | 51.21 | 51.11 |

the initial 2-4 rounds and then gradually decline due to overfitting. Thus, we use 3 global rounds in all of the multi-round experiments.

## D  PROOF OF THEOREM 1

According to Eq. 7 and Eq. 8, ignoring the learning rates, the difference of the global model can be bounded by:

$$\varepsilon \leq \sum_{i=1}^{m} \sum_{j=k+1}^{Tk} [(g_i(\boldsymbol{w}_i^{(0,j)}) - g_i(\boldsymbol{w}_i^{(t,j-kt)}))], \tag{10}$$

Considering Assumption 1, we have:

$$\varepsilon \leq \sum_{j=k+1}^{Tk} Lm\|(\boldsymbol{w}_i^{(0,j)} - \boldsymbol{w}_i^{(t,j-kt)}\|, \tag{11}$$

According to Assumption 2, we can deduce:

$$\varepsilon \leq \sum_{j=k+1}^{Tk} L\tau m\|\boldsymbol{w}^{(0,0)}\|, \tag{12}$$

Thus we have:

$$\varepsilon \leq L\tau Tkm\|\boldsymbol{w}^{(0,0)}\|, \tag{13}$$

which is Theorem 1.

## E  LIMITATION

This work has two main limitations. (1) The paper is limited in federated fine-tuning tasks since we lack the computation resources to conduct federated pre-training experiments. (2) Since common stable diffusion models do not vary significantly in parameter size, this work does not observe the performance of different-sized stable diffusion models in one-shot federated fine-tuning. The impact of model parameter size on one-shot federated fine-tuning in text-to-image generation tasks still needs to be explored.

