# OpenReview forum: "One Communication Round is All It Needs for Federated Fine-Tuning Foundation Models"
_ICLR.cc/2025/Conference — ICLR 2025 Conference Withdrawn Submission_

### Official Review · Reviewer_hfax · 2024-10-21

**Soundness:** 2
**Presentation:** 2
**Contribution:** 1
**Rating:** 3
**Confidence:** 5

**Summary:**

This paper studies the behavior of one-shot federated fine-tuning of foundation models (FM). In particular, it seeks to address the massive communication bottleneck when training foundation models in FL. Additionally, the paper claims that one-shot FL for FMs attain comparable performance to those achieved by multi-round FL, which deviates from the dynamics of smaller-scale models. The authors argue that their "breakthrough paves the way for broader adoption and application of federated fine-tuning" and that "a single communication round is all it needs to effectively fine-tune FMs".

**Strengths:**

Thanks for the interesting paper.

(1) I really like the related works section. It is clean, clear, and a pleasure to read.

(2) Originality and Significance: This paper claims to be the first work to report one round is sufficient for fine-tuning FMs. If valid, this would be a good contribution. The paper also attempts to draw a link between the "smoothness" of FM optimization and the reported phenomenon, which is interesting.

(3) The paper is well-written and easy to understand.

**Weaknesses:**

(1) Let's start with Figure 1, which evaluates ResNet-18 on CIFAR-10. The multi-round FL accuracy is <70 %. I find this concerning in a Cross-Silo setting. Additionally, more details would be needed for reproducibility. What hyperparameters did you use? How was the dataset partitioning done, as CIFAR-10 is a centralized dataset? How many clients are there? And how did you select the hyperparameters? Without this information, this figure is very difficult to interpret.

(2) The "one-shot" paradigm the paper studies is just a single round of FedAvg. Additionally, FedAvg is used as the only multi-round baseline, which is very problematic. There are very many algorithms that have been developed since FedAvg, and almost all works report FedAvg having lower performance. On a more foundational note, it is a bad idea to use FedAvg to train modern large-scale models, as that would correspond to LLM/Transformer training with local SGD in a distributed fashion. SGD is not the optimizer of choice for training such models, c.f., [1-3]. For example, the authors have cited [4], which already presents plenty of algorithms which makes minimal modifications to, but does significantly better than, FedAvg.

(3) My concern is more foundational. Clearly, the one-shot paradigm that the authors report is just a single step of FL/FedAvg (which is not the case for other papers, e.g., the ones cited in their literature review in Appendix). Given that the proof of Theorem 1 is 4 lines of straightforward equations, the central contribution of the paper comes from the discovery that "a single communication round is all it needs to effectively fine-tune FMs". The one-shot paradigm the authors consider is just a single step of standard FL training. Is this really a notable contribution? If so, wouldn't it be possible to argue that "two communication rounds do even better to effectively fine-tune FMs than a single communication round"?

(4) This paper misses comparisons with SoTAs, or even more modern distributed training algorithms other than FedAvg for that matter (e.g., [4]). The results of the paper show that even when FedAvg is used, one-shot is "comparable", which indicates that when properly trained (e.g., DiLoCo [3,5]), the one-shot approach studied in the paper will fare much worse. Additionally, the paper does not compare with any other one-shot approaches, which would confer the same communication complexity benefits as their single round of FedAvg.

(5) Where does equation (7) come from? If it's subtracting (5) and (6), don't we need that the local learning rate is fixed for this to hold? Additionally, the authors note that "The poor performance of one-shot FL is due to the gradients being calculated on the local models being less accurate than those calculated on the aggregated global model". One reason for this is due to client data heterogeneity, which induces local drift [6]. This causes issues with one-shot approaches in realistic, heterogeneous non-IID data settings that the authors note. Given that the proposal of the authors is FedAvg, there will be a point of diminishing returns as the number of local update steps k increase due to unmitigated client drift.

(6) It seems inaccurate to call Theorem 1 a theorem. The result follows immediately from computing the difference between equations (5) and (6) (difference between 1-shot model updates and multi-round model updates) and using the triangle inequality. And of course, the smaller model updates are assumed to be (Assumption 2), the differences between the one-shot and multi-round will be smaller. Also, if you lessen the number of update steps T, k, the difference is smaller between the 1-shot model and multi-round model as the authors restrict to the same number of update steps (Tk). Using this tautology to argue the validity of experiments is not convincing.

(7) In lines 248-249, the authors randomly sample a mini-batch of data to estimate the gradients. This needs a lot more clarification, as minibatch gradients are known to follow a heavy-tailed distribution for transformer-based models (e.g., [1-2]). In particular, this means that the L values empirically reported will be very noisy. Have averages been taken to form the L? Is code available to the reviewers for reproducibility of Figures? What is the batch size? How valid is the minibatch approximation, as there have been papers specifically studying the heavy-tailed phenomenon on modern models, which will significantly destabilize the validity of the approximation made by the authors?

(8) In Appendix B, why are the hyperparameters selected for the main experiments the optimal hyperparameters? How did the authors settle on choosing the learning rates?

(9) In Appendix C, the authors present a experiment which seems to show that model performance declines with more global training rounds. This may suggest that the hyperparameters selected is not optimal, or that centralized settings are more appropriate to train said model.

(10) Why does it make sense to make the restriction that the one-shot and multi-round FL should have the same kT? This is not fair at all, and would force multi-round to take shallower local updates. Another concern is that because the methodological contribution is just running one round of FedAvg, making that artificial restriction is the only way to possibly argue a contribution. Are we trying to search for the best performing model of between the training paradigms? If not, what are we trying to do, given that 1-shot approaches already exist?

----- References:


[1] Why are Adaptive Methods Good for Attention Models? (Zhang et al., NeurIPS 2020)

[2] Linear attention is (maybe) all you need (to understand Transformer optimization) (Ahn et al., ICLR 2024)

[3] DiLoCo: Distributed Low-Communication Training of Language Models (Douillard et al., Arxiv 2023)

[4] Adaptive Federated Optimization (Reddi et al., ICLR 2021)

[5] Asynchronous Local-SGD Training for Language Modeling (Liu et al., ICML workshop 2024)

[6] Breaking the centralized barrier for cross-device federated learning (Karimireddy et al., NeurIPS 2021)

**Questions:**

Please see the "Weaknesses" section. Here, I'll just list some less important concerns as my main questions are listed above. What I write here is probably not worth responding to.

Less important concerns:

-In all pseudogradient equations (3/5/6), there might be a typo-it looks like the right hand side is missing a "-".

-It is better to cite the published paper than an Arxiv version, e.g, Adaptive federated optimization. arXiv preprint (Reddi) has been published at ICLR.

-I feel the framing of the paper can be different. In my subjective opinion, the provided empirical justification of Assumptions 1 & 2 for FMs are a more surprising contribution.

-A less specific question concerns the practicability of this work. The reduction in communication by one-shot FL is indeed significant, and as the authors note in line 88, it may be a "game-changer for devices with limited bandwidth". I'm curious in general of why engineers would be fine-tuning foundation models on devices with limited bandwidth. A more thorough context of the necessity of this approach would strengthen the flow of the paper. I should note that this is not an important critique of the work.

---

### Official Review · Reviewer_JyeF · 2024-10-31

**Soundness:** 2
**Presentation:** 3
**Contribution:** 2
**Rating:** 5
**Confidence:** 4

**Summary:**

In contrast to traditional one-shot Federated Learning, the authors show that one-shot Federated Fine-Tuning (FT) is effective, in the sense that it can match the performance of multi-round Federated FT. This phenomenon becomes more prominent with larger Foundation Models (FMs). The authors analyze the equivalence by considering the final model difference between one-shot and multi-round FL and indicate key factors enabling it. Experiments on several FMs and 3 tasks demonstrate this performance match.

**Strengths:**

- This paper is well-written and addresses a relevant and important problem of Federated FT of LLMs.
- The observations made therein are interesting and could be useful to the wider Federated Learning community.
- The supporting theoretical analysis provides a reasonable justification in understanding the observed phenomenon.

**Weaknesses:**

Despite the interesting results, the paper is incomplete on a few fronts in its current form. I am listing them below:

1) It is unclear why the authors compare FL and one-shot FL (OFL) under the same number of total local steps (Tk). If the authors claim a similarity of performance, the same should be shown for the best FL vs. the best OFL i.e. under no constraints of local steps whatsoever. It could well be the case that the optimal parameter settings are different between the two.

2) The current experimental testbed is limited to showcase the generality of the claims. Do the observations hold for any level of heterogeneity? Are the observations consistent across a large number of clients (> 100)? It’s well-known that the performance of one-shot FL decreases with more clients. Hence, without more experiments across dimensions of heterogeneity and number of clients, the claims seem too bold. The number of clients appears as a key parameter influencing the bound in equation (9).

3) The zero-shot performance is fairly high. Thus the improvement sought by federated FT seems low. In some cases, FT also seems to deteriorate performance. Llama-13b achieves a zero-shot performance of 51.79 on ARC while the best Full FT (multi-round or one-shot, any dataset) is 47.57. The same seems to hold for Llama-7b on ARC where the zero-shot performance is higher than Full FT for both one-shot and multi-round, regardless of the training dataset (MMLU, Wizard, M-W). This raises questions regarding the suitability of the chosen datasets for supporting the claims.

4) The paper can benefit from the usage of more scientific language than words like “revolutionize”, “breakthrough”, etc. Even if large FMs are claimed to need only one communication-round, the required memory for training them on FL devices can prohibit their usage. In such cases, smaller models with multi-round training seem more viable.

5) The discussion in Section 5 is simply restating what is well-known within the one-shot FL community. It does not provide new insights. The reviewer recommends instead using the space for more ablation experiments and the experiments described above.

**Questions:**

1) What happens if you use more frequent communication i.e. local steps less than 1 epoch for multi-round FL? Ignoring the increase in communication costs, does it give better accuracy?
2) What happens if you use SOTA FL algorithms for multi-round FL? Is the performance of the one-shot vs multi-round still similar?

---

### Official Review · Reviewer_1Tgx · 2024-11-03

**Soundness:** 2
**Presentation:** 3
**Contribution:** 2
**Rating:** 5
**Confidence:** 4

**Summary:**

This paper presents empirical observations suggesting that large foundation models perform only marginal updates during the fine-tuning process. These observations are supported by a theoretical bound derived in this paper. The authors present extensive numerical experiments to support the claim that "for large FMs, one-shot federated fine-tuning can effectively replace multi-round federated fine-tuning." Subsequently, the benefits to federated learning are discussed in terms of communication efficiency.

**Strengths:**

- The paper makes many interesting numerical observations from various models, including model changes $\tau$, smoothness $L$, and the number of training steps $Tk$ required.
- A simple analytical bound is provided for the difference between one-shot FL and multi-round FL, offering insight into the empirical observations.
- The paper discusses the benefits of the observations to FL, highlighting improvements in communication efficiency.

**Weaknesses:**

- While the one-shot performances reported in the paper are "comparable" to the multi-round approach, it is challenging to conclude definitively that one-shot fine-tuning can fully replace multi-round fine-tuning. Please see question #1 below.
- The trade-off between performance and communication cost when one-shot federated fine-tuning may not outperform the multi-round method is overlooked in the paper. The paper could be improved by providing a quantitative analysis of the performance-communication trade-off, perhaps through a figure showing performance vs. communication cost for different model sizes, fine-tuning approaches, and number of local updates.
- The limitations need to be elaborated and clarified. It may be better to move this discussion to a dedicated discussion section for better visibility.
- It is unclear how these observations can be generalized to other models and under which settings they hold true. The paper could benefit from additional experiments by testing their approach on different types of foundation models (e.g., multimodal models, domain-specific models) or on fine-tuning tasks that are significantly different from the pre-training distribution.
- The paper is marginally relevant to FL, but could focus more on it and thus be strengthened by reporting, as also suggested above, a figure showing performance vs. communication cost for different model sizes, fine-tuning approaches, and number of local updates.

**Questions:**

1. The empirical observations are specific to LLMs and fine-tuning experiments, likely on similar datasets (in distribution). Can the main claim of this paper be generalized to other foundation models and fine-tuning experiments? For example, a similar observation was made in [1], where LoRA with rank 1 outperformed higher ranks, which should not be expected when downstream tasks are in a different language (i.e., different from pre-training). Therefore, it is also expected that one communication may not be sufficient when the fine-tuning task differs from pre-training. Could the authors discuss this?
2. Could the sequential aggregation result in Figure 5 be affected by the order of client arrivals? It should generally be influenced. Could the authors validate this by taking different orders of client arrivals?
3. In Assumption 2, how is $\tau < 1$ derived? Is this bound required for the proof?
4. The x-axis labels were cropped in Figure 2(a) and 2(b). Please correct them.
5. "learning rate to 8" in line 671 must be "batch size."
6. The proof of Theorem 1 may have error(s). Is (11) supposed to hold for all $i$? or is a summation missing?

**Reference**
1. Hu et al. LoRA: Low-Rank Adaptation of Large Language Models. ICLR 2022

---

### Official Review · Reviewer_7Xis · 2024-11-04

**Soundness:** 3
**Presentation:** 3
**Contribution:** 3
**Rating:** 5
**Confidence:** 4

**Summary:**

This paper stresses the challenge of the high communication cost in federated fine-tuning for foundation models. Toward the challenge, the paper proposed the one-shot federated learning framework to reduce communication costs, enhance privacy, and maintain performance consistency with multi-round federated fine-tuning.

**Strengths:**

1. This paper proposed the one-shot federated learning framework for fine-tuning foundation models.
2. The authors theoretically analyze the difference between one-shot FL and multi-round FL.
3. Experimentally, they show that large models have a higher level of smoothness, and require fewer fine-tuning steps, resulting in high model performance for one-shot FL compared to small models. The finding is interesting.
4. Experimental results validate the effectiveness of the proposed method.

**Weaknesses:**

1. The whole one-shot federated fine-tuning algorithm lacks an algorithm description. As there are many one-shot algorithms, which one is taken should be pointed out. Specifically, where is the foundation model, and what does the server do in the algorithm?
2. There are a global learning rate and a local learning rate in the formulation, as shown in (2) and (3). However, this is wired, as (2) seems a simple average operation.
3. The operation and number of training rounds of the server should be considered, which however is ignord in this paper.
4. The reasonability behind Assumption 2 should be verified. Is there any reference to support the assumption?

**Questions:**

Please refer to the Weakness.

---

### Note · Authors · 2024-12-03

I have read and agree with the venue's withdrawal policy on behalf of myself and my co-authors.